

# Assessment of online water-soluble brown carbon measuring systems for aircraft sampling

Linghan Zeng[1], Amy P. Sullivan[2], Rebecca A. Washenfelder[3], Jack Dibb[4], Eric Scheuer[4], Teresa L. Campos[5], Joseph M. Katich[3,6], Ezra J. T. Levin[2,7], Michael A. Robinson[3,6,8], and Rodney J. Weber[1]*

[1]School of Earth and Atmospheric Sciences, Georgia Institute of Technology, Atlanta, GA, USA
[2]Department of Atmospheric Science, Colorado State University, Fort Collins, Colorado, USA
[3]Chemical Sciences Laboratory, National Oceanic and Atmospheric Administration, Boulder, CO, USA
[4]College of Engineering and Physical Sciences, University of New Hampshire, Durham, New Hampshire, USA
[5]Atmospheric Chemistry Observations and Modelling Laboratory, National Center for Atmospheric Research, Boulder, CO, USA
[6]Cooperative Institute for Research in Environmental Sciences, University of Colorado Boulder, Boulder, CO, USA
[7]Handix Scientific, Boulder, CO, USA
[8]Department of Chemistry, University of Colorado Boulder, Boulder, CO, USA

*Correspondence to*: Rodney J. Weber (*rweber@eas.gatech.edu*)

**Abstract.** Brown carbon (BrC) consists of particulate organic species that preferentially absorb light at visible and ultraviolet wavelengths. Ambient studies show that as a component of aerosol particles, BrC affects photochemical reaction rates and regional to global climate. Some organic chromophores are especially toxic linking BrC to adverse health effects. The lack of direct measurements of BrC has limited our understanding of its prevalence, sources, evolution, and impacts. We describe the first direct, online measurements of water-soluble BrC on research aircraft by three separate instruments. Each instrument measured light absorption over a broad wavelength range using a liquid waveguide capillary cell (LWCC) and grating spectrometer, with particles collected into water by a Particle-into-Liquid Sampler (CSU PILS-LWCC and NOAA PILS-LWCC) or a mist chamber (MC-LWCC). The instruments were deployed on the NSF C-130 aircraft during WE-CAN 2018 as well as the NASA DC-8 and the NOAA Twin Otter aircraft during FIREX-AQ 2019, where they sampled fresh and moderately aged wildfire plumes. Here, we describe the instruments, calibrations, data analysis, and corrections for baseline drift and hysteresis. Detection limits (3σ) at 365 nm were 1.53 Mm$^{-1}$ (MC-LWCC; 2.5 min sampling time), 0.89 Mm$^{-1}$ (CSU PILS-LWCC; 30 s sampling time), and 0.03 Mm$^{-1}$ (NOAA PILS-LWCC; 30 s sampling time). Measurement uncertainties were 28% (MC-LWCC), 12% (CSU PILS-LWCC), and 11% (NOAA PILS-LWCC). The MC-LWCC system agreed well with offline measurements from filter samples, with a slope of 0.91 and R$^2$=0.89. Overall, these instruments provide soluble BrC measurements with specificity and geographical coverage that is unavailable by other methods, but their sensitivity and time resolution can be challenging for aircraft studies where large and rapid changes in BrC concentrations may be encountered.



# 1. Introduction

## 1.1 Importance of Brown Carbon

Organic compounds are a major component of ambient aerosol that affect atmospheric visibility, Earth's radiation balance and human health. In the past, all organic aerosol (OA) compounds were assumed to only scatter light and exert a cooling effect (Koch et al., 2007; Myhre et al., 2008). Recent studies have shown that a fraction of OA absorbs light with a strong wavelength dependence (Andreae and Gelencsér, 2006; Lack and Cappa, 2010). These absorbing OA components are referred to as brown carbon (BrC) because they have a brown or yellow appearance when concentrated, resulting from higher absorption at shorter

visible and ultraviolet (UV) wavelengths. This absorption offsets some portion of the scattering by OA. Modelling studies have suggested a non-negligible influence by BrC (Feng et al., 2013; Saleh et al., 2015; Zhang et al., 2017; Zhang et al., 2020). Actual global measurements of BrC, using the analytical methods discussed here, have shown that BrC can contribute up to 48% of the overall warming effect globally by absorbing carbonaceous aerosols (i.e., BrC + black carbon (BC)) (Zeng et al., 2020). Due to its absorption at UV wavelengths, BrC may also suppress photolysis rates of some chemical reactions, such as

decreasing surface ozone concentrations in certain locations (Jo et al., 2016). A fraction of BrC chromophores are composed of nitro- or oxy-aromatic species (Desyaterik et al., 2013; Zhang et al., 2013), which are known toxins (Bandowe and Meusel, 2017; Tian et al., 2020), making measurements of BrC chromophores a useful tool for assessing aerosol health impacts from specific emissions (Verma et al., 2015; Gao et al., 2020a; Gao et al., 2020b). Unfortunately, ambient observations of BrC have been sparse, limiting an assessment of its impacts and the refinement of model simulations.

## 50 1.2 Methods of Brown Carbon Measurement

Methods to determine BrC in suspended aerosol particles can be challenging. BrC and BC are often co-emitted, and must be distinguished by their unique properties, including (1) the wavelength-dependence of their absorption; (2) volatility; or (3) solubility.

*Brown Carbon Determined from Wavelength-Dependence of Absorption*

The strong wavelength-dependence of BrC absorption allows it to be determined from total absorption measurements (BC + BrC) at multiple wavelengths in some cases. This requires the assumption that BrC does not absorb at mid-visible and longer wavelengths and that the absorption Ångström exponent (AAE; Absorption~$\lambda^{-AAE}$) for BC is known and constant with wavelength. AAE for BC is commonly calculated by fitting the absorption measurement based on two wavelengths in the visible wavelength range or it is simply assumed to be one. BrC absorption at shorter wavelengths is then found by difference

from the extrapolated BC AAE (Lack and Langridge, 2013; Mohr et al., 2013). This approach can be applied to any technique that measures total absorption or absorption aerosol optical depth (AAOD) at multiple wavelengths, including filter-based methods, photoacoustic spectroscopy, and remote sensing, although what is measured as BrC is operationally defined by the measurement method. Filter-based absorption measurements have existed for some time (Lin et al., 1973), and may suffer from artifacts (Bond et al., 1999; Subramanian et al., 2007; Lack et al., 2008), although correction methods have been proposed



(Weingartner et al., 2003; Virkkula, 2010; Olson et al., 2015). Photoacoustic absorption spectroscopy measures aerosol light absorption at near-ambient conditions by heating particles from a controlled light source and detecting the soundwave, but is subject to interference by gaseous absorbers and sensitive to variations in temperature, pressure, and relative humidity (Arnott et al., 1999; Langridge et al., 2013). Ground-based remote sensing can determine AAOD at multiple wavelengths (Aerosol Robotic Network, AERONET; (Holben et al., 1998; Wang et al., 2016)). For each of these approaches, the AAE fitting and

extrapolation introduce uncertainties, including the calculation of AAE from only a few wavelengths (typically two) and the extrapolation to shorter wavelengths to determine a relatively small BrC contribution by difference. Studies that use an assumed AAE value introduce even greater uncertainty into the determination of BrC, since a range of values from 0.6-1.9 has been observed (Bergstrom et al., 2007; Bond et al., 2013; Lan et al., 2013; Li et al., 2016).

*Brown Carbon Determined from Volatility*

BrC may also be determined from total absorption measurements of thermally denuded and ambient samples (Cappa et al., 2012; Lack et al., 2012). The low volatility of BC means that it remains after thermodenuding, and the difference between the total absorption and denuded absorption can be used to determined BrC absorption. Separating BrC and BC absorption using either thermodenuding or wavelength-dependence can be complicated by morphological conditions, particularly the coating of BrC onto BC that results in increased absorption through lensing (Jacobson, 2001; Schnaiter et al., 2005; Bond and

Bergstrom, 2006). BC absorption can be enhanced due to a lensing effect involving an absorbing core covered by a scattering or slightly absorbing shell (Bond and Bergstrom, 2006; Cappa et al., 2012; Lack and Langridge, 2013), but a simple core-shell structure may not accurately represent the actual particle morphology, leading to further uncertainty (Sedlacek et al., 2012). Other studies suggest this enhancement is small in certain regions (Cappa et al., 2012; Cappa et al., 2019).

*Brown Carbon Determined from Solubility*

Finally, BrC may be determined by extracting BrC chromophores in solvents to separate them from insoluble BC, and measuring light absorption caused by the soluble organic chromophores (Hecobian et al., 2010). This is the only method to directly separate and quantify BrC. A spectrophotometer with a UV-Vis light source and long-path liquid waveguide capillary cell (LWCC) provide high spectral resolution and high sensitivity absorption measurements over a broad wavelength range through the use of long optical pathlengths. Direct measurement of organic chromophores is also useful for studying the

prevalence and fundamental properties of BrC, such as the impact of aging on optical properties and the toxicity of chromophoric species. However there are major limitations when using this method to determine aerosol optical effects since all particle size and morphological information are lost (Liu et al., 2013) and any BrC species insoluble in the selected solvents are not included. Other limitations include pH dependent absorption, blank stability, especially when using organic solvents, and artifacts which may be introduced by extensive dilution, resulting in changes in chemical properties of chromophores

relative to those of the ambient aerosol (Hinrichs et al., 2016; Phillips et al., 2017; Teich et al., 2017).

Spectrophotometric measurements of chromophores in solutions can be utilized in both offline and online systems (Hecobian et al., 2010; Liu et al., 2013; Zhang et al., 2013; Liu et al., 2015). For offline systems, atmospheric particles are usually collected by filtration, and then extracted with a solvent, such as water, methanol, or acetonitrile (Chen and Bond, 2010), and absorptions





of solvent and solute are quantified. Particle collection over a period of time onto a filter, followed by offline analyses at a
later date, can lead to artifacts through filter sampling biases and changes during storage. Low time resolution and the resulting
fewer data points can limit data interpretation. Among these weaknesses, poor time resolution is the most serious, especially
when sampling fire plumes using a fast-moving aircraft. Online measurements can improve this, but these have only been used
to measure water-soluble BrC due to the particle collection methods utilized. An online water-soluble BrC measuring system
with a Particle-into-Liquid Sampler (PILS)-liquid waveguide capillary cell (LWCC), has been used in previous ground-based
studies (Hecobian et al., 2010; Washenfelder et al., 2015). Other systems can be used to collect the aerosol into water for
subsequent BrC analysis, such as mist chambers (Cofer et al., 1985).

*Complications Due to Intermediate Compounds*

Separating BrC and BC by the wavelength-dependence of their absorption, their volatility, or their solubility is complicated
by the possible existence of compounds with intermediate properties of absorption, volatility, or solubility. Some studies show
evidence for the existence of intermediate BrC species, with properties between BC and BrC with a range of AAE values
(Saleh et al., 2018). This intermediate BrC has been suggested to be an incompletely pyrolyzed precursor to BC that shows
characteristics of both BC and BrC (Adler et al., 2019), much like what has been referred to as tar balls (Pósfai et al., 2004;
Chakrabarty et al., 2010; Adachi et al., 2019) and consistent with the idea that carbonaceous light absorbing aerosol is
comprised of a continuum of species from brown to black light absorbers (Cheng et al., 2021).

**1.3 This Work**

Here, we assess three systems for measuring water-soluble BrC (WS BrC) using either a mist chamber (MC) or PILS as the
aerosol sampling systems, followed by a LWCC and spectrometer (MC-LWCC and PILS-LWCC). These instruments were
deployed in three separate aircraft studies of wildfire smoke. MC sampling has been used in past NASA aircraft studies (Talbot
et al., 1999; Dibb et al., 2003; Scheuer et al., 2003), whereas this paper describes the first deployment of the MC-LWCC
system. Similarly, PILS-LWCC instruments have been developed and used in ground-based studies (Hecobian et al., 2010;
Washenfelder et al., 2015) and PILS systems have been deployed on aircraft to measure aerosol composition (Sullivan et al.,
2006; Sullivan et al., 2014; Sullivan et al., 2019), but this paper describes the first aircraft deployment of PILS-LWCC.

**2. Experimental**

**2.1 Overview of aircraft studies and brown carbon instruments**

The Fire Influence on Regional to Global Environments Experiment - Air Quality (FIREX-AQ 2019;
https://doi.org/10.5067/suborbital/firexaq2019/data001) and the Western Wildfire Experiment For Cloud Chemistry, Aerosol
Absorption And Nitrogen (WE-CAN 2018; https://www.eol.ucar.edu/field_projects/we-can) field studies investigated the
emissions and evolution of gases and aerosols from wildfires and prescribed burning to better understand fire impacts on air
quality and climate. FIREX-AQ included the NASA DC-8 research aircraft (average aircraft speed of 200 m/s), which was



deployed from Boise, ID and Salina, KS, USA during 22 Jul 2019 – 5 Sep 2019 and the NOAA Twin Otter research aircraft (average aircraft speed of 75 m/s), which was deployed from Boise, ID and Cedar City, UT, USA during 3 Aug 2019 – 5 Sep 2019. WE-CAN included the NSF C-130 research aircraft (average aircraft speed of 100 m/s), which was deployed from Boise, ID, USA during 22 Jul 2018 – 14 Sep 2018. For each campaign, large wildfires in the western United States were identified and flight plans included repeated plume intercepts to measure the smoke evolution.

The NASA DC-8, NOAA Twin Otter, and NSF C-130 payloads each included an instrument to measure WS BrC. These instruments employed similar approaches, but were developed separately. Briefly, ambient air was sampled through an aircraft inlet and then collected in aqueous solution using either a MC or PILS. The visible and ultraviolet absorption by the aqueous solution was determined using a deuterium/halogen lamp, LWCC, and grating spectrometer. All of the BrC measurements described in this work, including offline filter sample measurements, represent water-soluble BrC. Any intermediate species

that exist between BrC and BC are likely to be only nominally soluble in water, and we treat the water-soluble BrC measurements here as being solely attributable to BrC. The three instruments are summarized in Table 1, and described in greater detail below.

**2.2 Online Mist Chamber measurements on the DC-8 aircraft during FIREX-AQ 2019**

A mist chamber-ion chromatograph (MC-IC) system has been deployed on the NASA DC-8 research aircraft in many previous

missions for measurement of nitric acid and ionic particle species for all particles with sizes up to nominally 1 μm (Scheuer et al., 2003). We used the existing MC as an aerosol collection method and added a spectrophotometer for online measurement of BrC, without altering the existing MC-IC measurement capabilities. An instrument schematic is shown in Fig. 1. The mist chamber (or Cofer Scrubber) has been extensively used to collect water-soluble gases or particles (Cofer et al., 1985; Cofer and Edahl, 1986; Spaulding et al., 2002). It must be operated vertically with the top of the mist chamber connected to a vacuum

pump. Sample air flows in from the bottom and enters the mist chamber through a tube with a nozzle at the tube exit that is situated near the center of the mist chamber and then air exits the chamber at the top. Within this air jet, created by the nozzle, is a capillary that extends to near the bottom of the mist chamber. The low pressure near the air jet draws water sitting in the bottom of the chamber up the capillary, which breaks up the water into many small droplets within the air jet. The droplets and jet create a fine and uniform mist throughout the chamber, which is maintained in the chamber by a hydrophobic Teflon

filter at the top of the mist chamber that limits most of the water from leaving with the sample air that is continually being drawn through the chamber (Cofer et al., 1985). Droplets impact on this filter and the walls, keeping all internal surfaces wet and draining to the liquid reservoir at the bottom, where it is continuously recycled through the jet during the sample collection period. After the sample collection period, liquid is removed from the chamber and analyzed. For the BrC measurement, this involves transferring the liquid sample via a syringe pump with an associated multiport selection valve to a 2.5 m long liquid

waveguide capillary cell (LWCC-3250; World Precision Instruments, Sarasota, FL, USA, internal volume of 0.625 mL). The LWCC was coupled to a deuterium/halogen light source with spectral output from 200–2500 nm (DH-mini Light Source, Ocean Optics, Dunedin, FL, USA) and spectrometer (FLAME-T-UV-VIS, Ocean Optics, Dunedin, FL, USA). Light





absorption was measured and recorded between nominally 300 nm and 900 nm. A particle filter with 0.22 μm pore size (Polypropylene, Tisch Scientific, North Bend, OH, USA) was installed in front of the LWCC to prevent the long capillary
from becoming clogged and to limit contributions of insoluble particles larger than 0.22 μm to the liquid absorption measurement. During the FIREX-AQ study, the liquid particle filter was replaced and rinsed with water at the beginning of every flight. The LWCC system was connected to the MC through a single channel on a multiport selection valve downstream of the syringe pump. A portion of the MC liquid sample was analyzed with an IC and the remaining liquid in the chamber was directed to the LWCC through this separate channel. In this configuration, it did not affect the performance of the pre-existing
IC system for water-soluble ion quantification. Since the particle collection with the mist chamber was operated in batch mode, two MCs with identical corresponding syringe pumps operated alternatively, one sampling while the other was offline and the liquid sample was undergoing analysis. For example, a typical sampling sequence was as follows. The first MC (MC1 in Fig. 1) was filled with 12 mL of water via the syringe pump, and then the valve before the vacuum pump switched to allow sampling of ambient air at ~50 SLPM in that chamber. After 150 s of sampling, this valve was switched to sample from MC2 which had
been flushed and contained 12 mL of water in preparation for sample collection. Now offline, the syringe pump for MC1 withdrew 6 mL of water and directed 3 mL through the 2.5 m LWCC. Aabsorbance spectra were recorded when sample pumping was completed, meaning that the liquid tubing/filter (green path in Fig 1) and the LWCC had been flushed by about 3 volumes prior to the absorption measurement and the flow was stopped. After analysis and removal of any remaining sample liquid from the offline chamber, MC1 was then cleaned by flushing with 10 mL of water, but with no air flow. MC1 was then
ready for the next cycle of sampling. Liquid sample lines were 0.76 mm ID peek tubing and the volume between the MC and LWCC was 0.5 mL. There was a 166 s total time lag between the beginning of the actual sampling and the time the light absorbance spectrum was recorded.

A reference spectrum of pure solvent (water) was generated at the beginning of every flight at every wavelength ($I_0(\lambda)$ in Eqn (1)), and the light absorbance ($log_{10}\left(\frac{I_0(\lambda)}{I(\lambda)}\right)$) was quantified by the spectrometer over the full spectrum. The integration time
of the spectrometer was usually less than 0.1 s to keep the intensity at 365 nm in the range of 25,000 to 30,000 counts (i.e., below saturation). An internal standard of known aqueous concentration of trifluoroacetic acid (7.5 ppbm TFA) was added to the water supplied to the MC to track any evaporative loss of water from the MC during sampling, which was monitored with the MC-IC system. TFA did not interfere with the absorption measurement in the 300 nm to 700 nm wavelength range.

**2.3 Online PILS measurements on the NSF C-130 aircraft during WE-CAN 2018**

Unlike the mist chamber system, the PILS is run in a continuous sampling mode. The instrument operates by condensing water vapor onto particles with a saturated steam flow, and then using a single jet inertial impactor to collect the droplets onto a vertical impaction plate that is continually washed with a constant diluent flow (Weber et al., 2001; Orsini et al., 2003). Compared to a mist chamber, the PILS uses smaller liquid volumes, a smaller sampling flow rate, and produces a continuous liquid sample flow. The Colorado State University (CSU) PILS-LWCC system was similar to the one used in previous ground-





based studies (Hecobian et al., 2010), and the schematic is shown in Fig. 2a. Ambient air was sampled with a Submicron Aerosol Inlet (SMAI) (Craig et al., 2013a; Craig et al., 2013b; Craig et al., 2014; Moharreri et al., 2014) and passed through a nonrotating Micro-orifice Uniform Deposit Impactor (MOUDI) with a 50% transmission efficiency at 1 μm (aerodynamic diameter) at 1 atmosphere ambient pressure (Marple et al., 1991). The total airflow of the PILS was 15 SLPM (volumetric flow was controlled by a critical orifice). An activated carbon parallel plate denuder (Eatough et al., 1993) was placed upstream of the PILS to remove organic gases. The sample air then mixed with saturated water vapor (steam) in the growth chamber, and all particles in the sample air nominally larger than 40 nm grew to a few microns in size, and were then collected by impaction. The impaction plate was continually washed with a pure water transport flow of 1.3 mL/min dictated by the needs of the Total Organic Carbon Analyzer (Sievers M9 Portable TOC Analyzer; SUEZ Water Technologies & Solutions, Trevose, PA, USA) that was placed after the LWCC to quantify water-soluble organic carbon (WSOC). The liquid sample obtained from the PILS was then passed through a blown-glass debubbler, resulting in a liquid sample free of air bubbles at a flowrate of 1.2 mL/min, which was then filtered by a 0.2 μm pore size PTFE liquid particle filter (Whatman plc, Maidstone, UK) to remove larger insoluble particles. The flow was then directed through a 2.5 m liquid waveguide capillary cell (LWCC-3250, World Precision Instruments, Sarasota, FL, USA) and TOC Analyzer for near real-time measurement of WS BrC and WSOC, respectively. The LWCC was coupled to a deuterium/halogen light source (DH-mini Light Source, Ocean Optics, Dunedin, FL, USA) and spectrometer (FLAME-T-UV-VIS, Ocean Optics, Dunedin, FL, USA), the same model LWCC and spectrophotometer as used with the MC-LWCC and offline filter sampling system discussed below.

All liquid sample lines were 0.51 mm ID PEEK tubing. The liquid handling for the flows to and from the impactor used two pairs of syringe pumps with 1 mL syringes operating in handshaking mode. This minimized contamination observed by peristaltic pumps in previous ground-based studies, and provided more precise flow control for aircraft sampling with rapid changes in ambient pressure. One pair of syringe pumps delivered the transport flow to the top of the PILS impactor and the other pair withdrew the sample out of the PILS and pushed it through the liquid filter and to the LWCC and TOC Analyzer. A peristaltic pump handled other liquid flows to operate the PILS (see Fig. 2a). Using syringe pumps to move sample liquid to the LWCC has a major disadvantage because it alters the relationship between sample collection and analysis in each syringe stroke; the first liquid into the syringe is the last out (and last sample in is the first out), assuming minimal mixing in the syringe. This results in a roughly 1-minute (50 s) loss in sample time resolution (volume of syringe is 1 mL, flowrate 1.2 mL/min). The LWCC internal volume of 0.625 mL and liquid flow rate of 1.2 mL/min means the light absorption measurement is also averaged over a 32 s interval. The absorbance spectrum was saved every 16 s. Because of these effects, the time resolution of this method was roughly 1 minute. Periodic background measurements were made by manually switching a valve upstream of the PILS to direct sample air through a Teflon filter for 10 min. A dilution factor of 1.33 was used to account for dilution from steam condensation.



## 2.4 Online PILS measurements by NOAA on the NOAA Twin Otter aircraft during FIREX-AQ 2019

The NOAA PILS-LWCC instrument was developed separately, but followed the approach first described in Hecobian et al. (2010) and is similar to the CSU PILS-LWCC system. The key differences of the NOAA PILS-LWCC system included: (1) a pressure-controlled aerosol inlet with constant PILS gas flow; (2) an automated valve and aerosol filter to record aerosol-free

background measurements in flight; (3) a five-channel peristaltic pump for all liquid flow transfer, including the sample lines in and out of the PILS impactor. These details and the full system are described below. The schematic of the NOAA PILS-LWCC is shown in Fig. 2b.

During FIREX-AQ, the NOAA PILS-LWCC sampled from a forward-facing, near-isokinetic inlet (Schwarz et al., 2006; Perring et al., 2013) on the NOAA Twin Otter aircraft. The inlet flow was actively pressure-controlled at 620 hPa using a flow

restriction, pressure controller, and scroll pump. The total inlet sample flow of 8.13 SLPM passed through an impactor (TE296, Tisch Environmental, Cleves, OH, USA) with a measured 50% cut point at 0.95 μm, and was then distributed to the aerosol instruments onboard the NOAA Twin Otter. The PILS-LWCC sampled the incoming aerosol flow through an automated valve (MDM-060DT; Hanbay Laboratory Automation, Pointe Claire, QC, CAN) with filter (116IL; Headline Filters Limited, Aylesford, Kent, UK) for periodic, automated measurements of the aerosol-free background that were performed for 6 min

every 1.5 h. A parallel-plate carbon filter denuder (DN-100; Sunset Laboratory, Tigard, OR, USA) removed gas-phase volatile organic compounds (VOCs). Pressure and temperature of the flow were recorded at 1 Hz.

The PILS (PILS 4001; Brechtel Manufacturing Inc., Hayward, CA, USA) collected aerosol in solution using a steam generator, droplet impactor, and five-channel peristaltic pump, with an average liquid output flow from the impactor of 1.53 mL/min. The sample air flow into the PILS was maintained at a constant 6.0 SLPM using a 1.35 mm diameter critical orifice (Lenox

Laser, Glen Arm, MD, USA) between the PILS and the scroll pump. The pressure-controlled inlet and constant gas flow had two advantages for the PILS-LWCC: (1) The steam temperature within the PILS varies with pressure according to the Clausius-Clapeyron equation, and maintaining a constant gas pressure within the PILS allowed more stable behavior and better characterization of the PILS collection efficiency; (2) The liquid flow from the peristaltic pumps was found to vary with the system pressure, and maintaining constant upstream pressure improved the system stability and accuracy. Bubbles were

removed using a commercially-available flow-through debubbler (Omnifit 006BT; Diba Industries, Inc., Danbury, CT; modified by Brechtel Manufacturing Inc) consisting of a porous PTFE membrane under vacuum.

Following the PILS, a particle filter with 0.2 μm pore size (Puradisc 25 TF; GE Healthcare Life Sciences, Pittsburgh, PA, USA) removed insoluble components from the liquid sample stream before it entered the 2.5 m-long LWCC, which is the same model used in the other instruments (LWCC-3250, World Precision Instruments, Sarasota, FL, USA). Subsequently, 1.1

mL/min was sampled by a TOC analyzer (M9 Portable TOC Analyzer; GE Analytical Instruments Inc., Boulder, CO, USA) for measurement of WSOC and the excess flow (~0.43 mL/min) was directed by an automated 14-port valve (C25Z; Vici Valco Instruments, Houston, TX, USA) to a series of 12 polypropylene sample tubes for offline analysis or to a waste container.



Similar to the CSU PILS-LWCC, the optical system consisted of a deuterium/halogen light source nm (DH-mini; Ocean Optics Inc., Dunedin, FL, USA) coupled to the LWCC, however in this case, the exiting light was coupled to a 101 mm focal length

symmetrical cross Czerny-Turner spectrometer with a 18-bit back-thinned 1024 × 58 pixel CCD array detector cooled to -5 deg C (QE Pro; Ocean Optics Inc., Dunedin, FL, USA). The spectrometer contained a 600 groove/mm grating (300 nm blaze wavelength) rotated to give a useful spectral range from 309 to 682 nm. The entrance slit was 200 μm wide × 1000 μm tall, and was illuminated by a fiber bundle containing a linear array of 200 μm diameter UV/Vis fibers. 50 spectra with 0.02 s integration time were averaged to 1 Hz and saved. Following the field campaign, the collection efficiency of the PILS system

at 620 hPa was measured using atomized sucrose aerosol, and the BrC absorption and WSOC concentrations were subsequently corrected by a factor of 1.25. Since this system used a peristaltic pump (in contrast to the syringe pump of the CSU PILS-LWCC system) to move liquid sample from the PILS to the LWCC, the time resolution should be improved.  At a liquid sample flow rate of 1.53 mL/min through the LWCC of internal volume 0.625, the maximum possible time resolution would be 25 s, assuming no other interferences.  The observed time resolution is 60 s, likely due to sample mixing at the PILS

impaction plate, liquid fittings, and other instrument components.

**2.5 Offline measurements from filter samples on the DC-8 aircraft during FIREX-AQ 2019**

Filters were also collected as part of the FIREX-AQ NASA DC-8 measurement suite and BrC was determined offline with the same analytical method used in previous missions (i.e., NASA SEAC$^4$RS, DC3, and ATom, which are described in detail elsewhere (Liu et al., 2014; Liu et al., 2015; Zhang et al., 2017; Zeng et al., 2020)). Atmospheric particles with aerodynamic

diameters less than nominally 4.1 μm were collected onto 90 mm diameter Teflon filters with 1 μm pore size (MilliporeSigma, Burlington, MA, USA) (McNaughton et al., 2007). During plume sampling, each filter sample was timed to (as best as possible) coincide with a transect through a single smoke plume. During other periods, sampling times were generally 5 min or less when sampling at altitudes below 3 km and increased to a maximum of 15 min for higher altitudes. Subsequently, the filters were extracted first into 15 mL of water via 30 min of sonication and then using a syringe pump, extracts were filtered and

injected into a 2.5m LWCC (LWCC-3250, World Precision Instruments, Sarasota, FL, USA), coupled with the same light source and spectrometer (USB-4000, similar to FLAME-T-UV-VIS, Ocean Optics, Dunedin, FL, USA). The air filters were dried passively and then extracted again in 15 mL of methanol, and this extraction liquid was filtered and injected with the syringe pump into the LWCC. Only the water extracts are discussed here for comparison to the MC-LWCC. The same type of 0.22 μm pore size particle filter as the online system was installed in front of the LWCC to filter out insoluble particles for

both the water and methanol extracts, and the particle filter was changed every 5 to 20 samples depending on the sample concentration. Overall, the spectrometer was operated in the same way as the online MC system. Some samples collected in thick fire plumes were diluted to prevent saturation of the raw absorbance signal. Due to high organic concentrations in the filter extracts, the waveguide required periodic cleaning. Contamination was observed as the signal intensity for pure solvent, $I_0(\lambda)$, decreasing as contaminates accumulated in the waveguide. Flushing the waveguide with a large volume (50 mL) of





water was generally sufficient to clean it, but occasionally a stronger cleanser (10% of Contrad-NF, Decon Labs, King of
       Prussia, PA, USA) was used, as recommended by the manufacturer.

**2.6 Calculation of light absorption for PILS-LWCC, MC-LWCC, and filter samples**

Light absorption by the liquid in the LWCC is described by Beer's Law:

$$Abs_{solution}(\lambda) = c_{solution} \cdot \sigma_{solution}(\lambda) = \frac{1}{l} \cdot log_{10}\left(\frac{I_0(\lambda)}{I(\lambda)}\right) \tag{1}$$

where $Abs_{solution}(\lambda)$ is the absorption of the solution, $c$ is concentration, $\sigma_{solution}(\lambda)$ is the mass absorption efficiency, $l$ is
       the LWCC cell length, $I_0(\lambda)$ is light intensity in the absence of the absorber, which is the spectrum of pure water, and $I(\lambda)$ is
       light intensity with the absorber present. This can be converted to an absorption coefficient for the aerosol:

$$Abs_{aerosol}(\lambda) = \frac{V_{solution} \times log_{10}\left(\frac{I_0(\lambda)}{I(\lambda)}\right)}{V_{air} \times l} \times ln(10) \tag{2}$$

       For online PILS-LWCC measurements, $V_{solution}$ is the liquid sample flow rate and $V_{air}$ is the air flow rate. For the MC-LWCC
and offline filter measurements, $V_{solution}$ is the liquid sample volume for extraction and $V_{air}$ is the sampled volume of air. The
       light absorption determined from Eqn (2) is not directly equivalent to the ambient particle light absorption coefficient due to
       Mie effects. To determine the absorption that would be observed in the particle phase, the solution-phase absorption must be
       corrected as described previously (Liu et al., 2013; Zeng et al., 2020). The correction can be calculated from the imaginary
       part ($k$) of the aerosol complex refractive index, $m = n + ik$, and the measured size distribution. In this work, we report only
the light absorption from water-soluble chromophores, calculated from Eqn (2).

       The absorption coefficient at 365 nm ($Abs_{365nm}$) is used to represent WS BrC absorption in the analysis and figures, and is
       determined by averaging from 360 nm to 370 nm. $Abs_{675nm}$ (670–680 nm average) or $Abs_{700nm}$ (695–705 nm average) is
       used as a baseline to monitor any air bubbles or insoluble BC passing through the liquid particle filter (pore -size: 0.22 μm for
       MC and filter; 0.2 μm for PILS) with the assumption BrC does not absorb light in these wavelength ranges. All absorption
coefficient data have been blank corrected by water blanks for MC-LWCC and field blanks for PILS-LWCC by switching the
       upstream valve. Data from the MC-LWCC and PILS-LWCC instruments were time-corrected due to delays in the liquid flow
       system between sampling and analysis.

**2.7 Other Measurements**

       Carbon monoxide (CO) is emitted by biomass burning and is relatively chemically inert. It is often used as a marker for smoke
and as a tracer for determining plume dilution. For the NASA DC-8 during FIREX-AQ, CO was measured with a diode laser
       spectrometer method (Differential Absorption Carbon Monoxide Measurements; DACOM; (Warner et al., 2010)). For the
       NSF C-130 during WE-CAN, CO was measured by a quantum cascade laser instrument (CS-108 miniQCL, Aerodyne
       Research Inc., Billerica, MA, USA). For the NOAA Twin Otter during FIREX-AQ, CO was measured by cavity ringdown
       spectroscopy (G2401-m; Picarro Inc., Santa Clara, CA, USA; (Crosson, 2008; Karion et al., 2013)).



Refractory Black Carbon (rBC, or just BC) was measured by a single particle soot photometer (SP2) on the DC-8 and the C-130 aircraft. The SP2 measures the incandescent signal generated from single particles heated by a laser source, which is proportional to their mass (Schwarz et al., 2008). The SP2 measured rBC particles with mass equivalent diameters between ~90–550 nm on the DC-8 during FIREX-AQ and ~90–500 nm on the C-130 during WE-CAN. Higher frequency CO and BC data were merged to the MC-LWCC, PILS-LWCC or filter collection times as needed.

**3. Results and Discussion**

**3.1 Brown Carbon Measurements in Smoke Plumes**

Example flight tracks downwind of wildfires are shown in Fig. 3 for the three aircraft. Each of these flights sampled a single fire complex, with initial transects close to the source, followed by a pattern of downwind transects ideally perpendicular to the dominant wind direction. This type of sampling was repeated for numerous fires throughout each field study. The

corresponding time series of $Abs_{365nm}$ and baseline $Abs_{700nm}$ ($Abs_{675nm}$ for NOAA PILS-LWCC) for these flights are shown in Fig. 4. 1 Hz CO concentrations are plotted to identify when the aircraft was in smoke and indicate smoke concentrations. The peak CO values decreased downwind as the plumes dispersed and diluted with cleaner background air as they advected away from the fire.

For these methods, in general, $Abs_{365nm}$ has a similar trend with CO, but there are discrepancies. For the NASA DC-8

sampling during FIREX-AQ, the typical transit time through the plume was ~3 min; the MC-LWCC sample time was 2.5 min, thus the sampling frequency with this instrument was not sufficient to resolve structure within the plume during one transect. In contrast, the PILS-LWCC on the NSF C-130 and NOAA Twin Otter provided better time resolution. The average time for both the NSF C-130 and NOAA Twin Otter to transit a plume was ~4 min since their air speeds were lower than the DC-8. The data for all three systems show an increase in baseline ($Abs_{700nm}$ or $Abs_{675nm}$) within the plumes and there is evidence

of hysteresis in the BrC measurements. Both of these issues are discussed next.

**3.2 Baseline drift correction using long-wavelength absorption**

The accuracy of $Abs(\lambda)$ calculated from Eqn (2) may be limited by drift in $I(\lambda)$. Potential sources of drift in $I(\lambda)$ include air bubbles in the LWCC, variable levels of insoluble BC that passed through the particle filter, or changes in the light source intensity. Measured absorption at visible wavelengths can be used as a correction for air bubbles or insoluble BC in the LWCC.

The presence of insoluble BC has been reported by Phillips and Smith (2017) for methanol extracts. They observed that the long-wavelength absorption decreased when filtering the liquid extract with smaller pore size particle filters. We use $Abs_{675nm}$ or $Abs_{700nm}$ for corrections here. This requires the assumption that WS BrC does not absorb at 675 or 700 nm.

*MC-LWCC:* Figure 4a shows that $Abs_{365nm}$ is correlated with CO, with some differences. $Abs_{700nm}$ is generally less than ~0.5 Mm$^{-1}$ when sampling background air, but increases with $Abs_{365nm}$ in smoke plumes. Air bubbles or small insoluble black

carbon particles that pass through the particle filter may lead to elevation of $Abs_{700nm}$ as well as $Abs_{365nm}$. The scatter plot





between $Abs_{365nm}$ and $Abs_{700nm}$ for all samples is shown in Fig. 5a. There are two groups of data. Blue data points are interpreted to be small air bubbles in the sample liquid that were introduced during the syringe pump valve switching or due to leaks at liquid sample-line joints. The result is an upward shift of the complete absorption spectrum, which is consistent with a regression slope between $Abs_{365nm}$ and $Abs_{700nm}$ of approximately 1, and this error can be corrected by subtracting

$Abs_{700nm}$ from $Abs(\lambda)$.

In most cases, however, the presence of $Abs_{700nm}$ is likely due to a fraction of BC, which absorbs light at higher wavelengths (e.g., 700 nm), that passed through the 0.22 μm particle filter. $Abs_{700nm}$ is found to have a good correlation (Fig. 5e, $R^2$=0.65) with BC mass concentration consistent with BC as a potential contributor of long wavelength absorption. These data suggest the MC-LWCC is at least somewhat effective at collection of insoluble species into water.

**Filters:** In contrast to the MC system, when using a filter as the particle collection method $Abs_{700nm}$ does not have any correlation with $Abs_{365nm}$, and the magnitude of $Abs_{700nm}$ is much smaller compared to that for the MC. The random interference at $Abs_{700nm}$ is not likely due to BC as $Abs_{700nm}$ is independent of the BC mass concentration, as shown in Fig. 5f. The filter collects insoluble particles, but apparently the water extraction process does not efficiently move these particles from the filter to the extraction water in comparison to the MC-LWCC.

**CSU and NOAA PILS-LWCC:** A slight, but much smaller increase in the light absorption at high wavelengths is seen in the CSU (Fig. 4b) and NOAA PILS-LWCC data (Fig. 4c). Similar scatter plots, Figs. 5c and 5d, were made for data from the CSU PILS-LWCC and the NOAA PILS-LWCC, where the data were classified by flight using different colors. For the CSU PILS-LWCC, the slope of $Abs_{365nm}$ to $Abs_{700nm}$ varies between flights, whereas for the NOAA PILS-LWCC a similar relationship was seen for different flights. Comparing the CSU PILS-LWCC $Abs_{700nm}$ light absorption in Fig. 5g shows in most flights

some relationship to BC, or no relationship for a few flights. (A similar plot is not included for the NOAA PILS since no BC data were available). The results suggest that some BC contributed to the PILS-LWCC measurement, but it was minor compared to the MC-LWCC (note, difference in axis scales). Previous studies have indicated that the PILS is not a good collector for water-insoluble species (Peltier et al., 2007). Since this analysis suggests it is largely due to some fraction of BC being included in the BrC measurement, this BC interference can be removed.

**Baseline Correction:**

$Abs(\lambda)$ can be corrected for absorption by insoluble BC by assuming that $AAE_{BC}$ = 1 and $Abs_{700nm}$ is due to BC with no contribution from BrC. These are the same assumptions that other optical instruments use to infer BrC from total light absorption, as described in the introduction. With the assumption that $Abs_{700nm}$ is solely due to BC, $Abs_{365nm}$ due to BC can be estimated to be equal to $(365/700)^{-1} \times Abs_{700nm}$. According to the slope of the red line in Fig. 5a, BC contributes to

about one third of $Abs_{365nm}$ measured with the MC. Alternatively, one can simply subtract the measured absorption at all wavelengths, including ($Abs_{365nm}$) by $Abs_{700nm}$ (BrC, or corrected $Abs_{365nm} = Abs_{365nm} - Abs_{700nm}$). This simplified method results in 25% overestimation of BrC for the MC data compared to estimating the contribution of BC as a function of wavelength (BC AAE=1). Therefore, in the following analysis for the MC-LWCC system, WS BrC was calculated by





$WS\ BrC = Abs_{365nm} - (365/700)^{-1} \times Abs_{700nm}$. For the CSU PILS-LWCC, the overestimation is between 2 and 5% as

BC is not as efficiently collected and transported to the LWCC. Thus, in the following only the simplified method is used to

correct for BC interference, that is CSU $WS\ BrC = Abs_{365nm} - Abs_{700nm}$. The same approach was used for the filter data.

No correction for BC in the NOAA PILS data is made because of an observed slow baseline drift for 365 nm and 675 nm

absorption, possibly caused by independent drifts in the output of the deuterium lamp (~200 - 400 nm) and halogen lamp (~400

-1600 nm) within the DH Mini light source. In any case, as noted above, the correction would be small (<5%).

**3.3 Hysteresis**

An effect of retention of liquid on the internal wetted components (Gomes et al., 1993) or within dead volumes (i.e., poorly

flushed volumes within fittings or components) in the instrument is an observed hysteresis, which appears as a tail or

asymmetry in measurement peaks toward larger times. As seen in Fig. 4, WS BrC ($Abs_{365nm}$) demonstrates hysteresis when

the aircraft exited a smoke plume for both the MC-LWCC and PILS-LWCC systems, whereas the CO mixing ratio decreases

sharply (i.e., any hysteresis associated with the CO measurement is much less). The hysteresis in WS BrC results in it being

overestimated when the aircraft moves out of a polluted region to a cleaner environment due to high residual concentrations

from the previous run. Conversely, when moving from a region of low to high concentration, such as entering the smoke

plume, cleaner sample liquid from the previous airmass sampled can dilute the current measurement, resulting in an

underestimation of the BrC levels. The hysteresis effect is most obvious when the plume concentration changes significantly

during a short period. These large hysteresis effects can to some extent be removed.

*MC-LWCC:* For the MC BrC measurement system, liquid remains in the MC, syringe, and liquid lines associated with a

specific MC (red in Fig. 1, this whole group is referred to as just the MC) and in the common sample line and liquid filter

shared by both MCs (green in Fig. 1) from the previous sample, although these common lines are flushed with sample prior to

the measurement to minimize some of the latter hysteresis effect. Some fraction of the hysteresis can be removed by estimating

its contribution based on comparison to a measurement that is not as affected by hysteresis, such as the CO measurement. The

approach is based on two assumptions: (1) The volume fraction due to residues from the previous run does not change, making

a constant hysteresis effect, no matter dilution or enrichment; (2) The WS BrC level is zero (or at least much lower compared

to in the plume) when the CO concentration is at background concentrations, in this case we assume this occurs if CO < 80

ppbv. For i-th MC sample, we decompose the hysteresis into two components: (1) residue from a previous run of the same MC

used to collect the currently analyzed sample, which, since there are two MCs running alternatively, is sample from the (i-2)-

th sample (i.e., red components in Fig. 1) and (2) residue from the tubing transporting the liquid from the MC to the LWCC

(including the particle filter) which comes from the (i-1)-th (i.e., immediately preceding) sample (green components in Fig.

1). We pick time periods when the DC-8 was exiting fire plumes, in which case the observed WS BrC signals following the

smoke plumes were ideally all due to contribution from the previous airmass sampled.



We assume $a\%$ of the observed WS BrC absorption at i-th sample is due to the real WS BrC during the time period of the i-th sample, $b\%$ due to (i-1)-th sample from the tubing, and $c\%$ due to (i-2)-th sample from the MC. Based on mass conservation, the relationship between the coefficients and absorption can be described by the following equations.

$$a + b + c = 100 \tag{3}$$

$$Abs_{i,observed} = a\% \times Abs_{i,real} + b\% \times Abs_{i-1,observed} + c\% \times Abs_{i-2,observed} \tag{4}$$

Rearranging Eqn (4),

$$a\% \times Abs_{i,real} = Abs_{i,observed} - b\% \times Abs_{i-1,observed} - c\% \times Abs_{i-2,observed}$$

Using the sample immediately after the DC-8 had just exited the plume and labelling the last WS BrC measurement within the plume as $Abs_{0,observed}$ the series of measurements can be described as:

$$a\% \times Abs_{2,real} = Abs_{2,observed} - b\% \times Abs_{1,observed} - c\% \times Abs_{0,observed}$$

$$a\% \times Abs_{3,real} = Abs_{3,observed} - b\% \times Abs_{2,observed} - c\% \times Abs_{1,observed}$$

$$a\% \times Abs_{4,real} = Abs_{4,observed} - b\% \times Abs_{3,observed} - c\% \times Abs_{2,observed}$$

…etc.

According to the second assumption, the left side of the equations are zero; the WS BrC levels outside the plume are zero (or much lower than those within the plume). These equations can be solved by least squares for an overdetermined system to

obtain the coefficients, which as noted, we assume are constant. For our MC setup we find: $a = 56 \pm 13$, $b = 7 \pm 4$, and $c = 37 \pm 12$ based on integrating the result from nine cases of plume exits, where the uncertainties are the standard deviations from multiple plume exit analyses. In other words, for one measurement of WS BrC absorption, about 56% of absorption is from the current i-th sample, and 7% from residue in the tubing due to (i-1)-th sample and 37% from MC residue due to (i-2)-th sample. The largest source is residue in the MC. Although 10 mL of water was used to clean the chamber, since there was

no airflow during the cleaning, no mist was generated and so the walls, nozzle and water-refluxing filter of the MC were not rinsed. As for hysteresis in the sample lines and LWCC, the length of the tubing between the MCs and LWCC was as short as possible (about 1 m long), but there was still 0.5 mL of liquid volume (with 0.76 mm ID tubing) for the instrument arrangement on the DC-8. The internal volume of the 2.5 m LWCC was 0.625 mL. In our setup, the maximum liquid available for the BrC measurement with the LWCC was 3 mL (the IC analysis required most of the MC liquid sample), thus based on the volume

of the sample line and LWCC combined, these components were roughly flushed twice with the sample and the third volume was used for the analysis (internal volume of tubing and LWCC was approximately 1 mL and volume of liquid sent through the system in each MC analysis was 3 mL). Flushing with more sample (i.e., use more water in the MC) would reduce the hysteresis, but the under-measurement of the peak BrC levels within the plume will remain and it will also reduce the sensitivity of the overall BrC measurement if all other factors, such as sampling time, remain the same. From these analyses, the greatest

improvement in the sampling system could be gained by minimizing hysteresis from residue in the MC from the previous sample.





Figure 6a shows the time series plot of WS BrC ($Abs_{365nm} - (365/700)^{-1} \times Abs_{700nm}$) with CO before and after applying the hysteresis correction. This figure shows that corrected WS BrC is higher within the plumes and lower outside the plumes and in better agreement with the CO trend, which is not affected by hysteresis. However, this method does not completely

remove all hysteresis as some disagreement still exists. The uncertainties of the measurements also increased as the hysteresis is not a constant; the uncertainties of the three factors *a*, *b*, and *c* are not insignificant, as assumed.

The effect of this correction can also be assessed through scatter plots of measured WS BrC and CO before and after the correction, as shown in Figs. 7a and 7b, respectively. A better correlation is observed when the hysteresis correction is applied; $R^2$ increases from 0.75 to 0.86.

***CSU and NOAA PILS-LWCC:*** A similar hysteresis of WS BrC is seen in the continuously flowing liquid system of the CSU PILS-LWCC relative to the CO data in Fig. 4b. Here we attempt to remove some of this effect by applying a similar analysis method used for the MC using the raw higher time resolution BrC data. Unlike the dual MC system, the hysteresis could occur from hang-up of liquid in the internal wetted components encompassing the PILS impaction plate to the LWCC, along with the mixing caused by using a syringe pump (i.e., first liquid in is last out, discussed above). To correct the hysteresis effect,

similar assumptions noted for the MC also apply here. In this case, we assume *d%* of the current absorbance observation ($Abs_{j,observed}$) is due to current j-th sample ($Abs_{j,real}$), and (1- *d%*) absorbance is due to hysteresis ($Abs_{j-1,observed}$). The relationship can be described as:

$$Abs_{j,observed} = d\% \times Abs_{j,real} + (1 - d\%) \times Abs_{j-1,observed} \tag{5}$$

We also assume that WS BrC is nearly zero when the CO mixing ratio is less than 80 ppbv, which means the first term on the

right side of Eqn (5) vanishes. Again, using data when the C-130 just exited the plume and then following with ten background samples (CO<80 ppbv, but where WS BrC is not zero due to the hysteresis effect), the overdetermined system was also solved with the least square method, but this time only with one unknown, *d*. The mean and standard deviation of the factor *d* with data from 10 plumes analyzed is 11 ± 2 (again, ± is the standard deviation in d determined from multiple plumes). Figure 6b shows the result. The trend of the corrected WS BrC corresponds better with the CO trend, but is noisier than the original data

due to the corrected WS BrC being derived by dividing by a small number, i.e., *d%*. The noise data may also be due to an overcorrection. The data shown in Fig. 4b is the most exaggerated case of hysteresis encountered during WE-CAN. When lower concentration smoke plumes, generally when CO was less than 2000 ppbv, the hysteresis effect was not apparent, indicating a correction may not be necessary in all cases. To remove the added noise, the data could be smoothed (over longer time intervals, 150 s was found to be optimal), but that will reduce the time resolution of the measurement. Similar to the MC

results, better correlation is seen in the scatter plots between WS BrC vs. CO, shown in Figs. 7c and 7d, where the $R^2$ increases from 0.49 to 0.58. The greater scatter for the PILS data compared to the MC is likely due to the C-130 in WE-CAN flying over more fires that were relatively small while the DC-8 focused on larger stronger plumes in FIREX-AQ. Therefore, the WE-CAN data was more influenced by variability in the WS BrC vs. CO between different smoke plumes. The hysteresis phenomenon (Fig. 3c) is not obvious for the NOAA PILS-LWCC compared with CSU PILS-LWCC (Fig. 3b), possibly due to

the smaller dead volume throughout the system (e.g., use of a inline bubble trap with PTFE membrane versus glass bulb



debubbler), the use of a peristaltic pump versus syringe pump to move sample liquid, avoiding the syringe pump sample mixing issues, and overall lower concentrated plumes being sampled by the Twin Otter (most smoke plumes had CO below 2000 ppbv compared to typical CO of more than 3000 ppbv for the plumes encountered in WE-CAN mission). Because of this, no overall hysteresis correction was performed for the NOAA PILS-LWCC.

### 485    3.4 Comparison between MC and Filter Measurements of BrC

We also compare the online WS BrC measurements to filter sample results, noting that the filter does not have this liquid hysteresis issue. This can only be done for the MC measurements of WS BrC since of the three aircraft, only the DC-8 included a particle filter sampling system allowing for off-line BrC aerosol particle analysis.

The MC-LWCC BrC data with 2.5 min resolution were averaged to the lower time resolved offline filter data. Figure 8 shows
the comparison between these two methods. In the comparisons that follow, all the MC-LWCC data have been corrected for the baseline drift likely due to BC, as discussed in Sect. 3.2. Fairly good agreement is found between the online and offline BrC measuring systems when not corrected for the hysteresis associated with the online data, with a slope of 0.74 and $R^2=0.84$. However, the agreement is better once the hysteresis correction is applied, with slope of 0.91 and $R^2=0.89$. The improvement in $R^2$ is less than that seen for the comparison with CO in Fig. 7 and is likely due to averaging the MC WS BrC data to the
longer filter sampling times and that most filter sampling times where restricted to periods within the smoke plumes (i.e., less data for periods of transition from within to outside of plumes). Overall, the agreement suggests that the filter measurement of BrC is not biased by possible sampling artifacts associated with absorption of gases or evaporative loss of BrC components from the filter, which is common for filter sampling of semi-volatile species, but not as significant an issue for online sampling systems, such as the MC-LWCC.

### 500    3.5 Detection limits and measurement uncertainty

The limit of detection (LOD) and measurement uncertainty of the three instruments are presented below. The detection limits depend on the spectral integration time, sample air flow rate, volume of extraction water, and also the optical path length of the waveguide.

*MC-LWCC:* The air sample flow rate directly affects the detection limit and thereby the sensitivity of the MC-LWCC for
measuring BrC. The nominal flow rate of the MC-LWCC was set to ~50 SLPM. For a given MC-LWCC design, a sufficient flow rate is necessary so that particles are efficiently scrubbed, and all internal surfaces are wetted and continually flushed. The highest flow rates possible are also limited by the pressure drop across the Teflon water-refluxing filter and vacuum pump size. Longer sampling times will have larger volumes of air sampled as well as cause more evaporation of the water in the MC-LWCC, both increase the sample concentration, but reduce the time resolution. The water level in the MC-LWCC should
be kept as low as possible to have the highest concentration, but must be sufficient to maintain a reservoir in the bottom at all times while operating so that a mist is continually maintained and all surfaces are wetted and drained during sampling. Also, there must be sufficient sample for the various measurements. Insufficient sample can lead to drawing in air bubbles into the





analytical instruments, which, depending on the amount, can invalidate the measurement or as seen, cause extensive baseline drift.

Based on experiments, the optimal sampling time and water injection volume were found to be 150 s and 12 mL for this MC and analytical system. Approximately 10 mL of liquid remained in the chamber after sampling was completed. 3 mL of ambient sample was the maximum volume that could be injected into the waveguide without interference to the IC system. Occasionally, some air bubbles were introduced into the waveguide due to insufficient liquid volume left in the MC or the system leaking, as discussed above. Two cycles of liquid injection were found to be enough in most cases to remove the

resulting absorption baseline drift caused by air bubbles.

Limits of detection (LOD) are typically calculated by three times the standard deviation of the blank measurement; however, we did not have any blank measurements involving filtering out aerosol in the MC-LWCC during the FIREX-AQ campaign. Instead, a water blank at the beginning of each flight was used, where pure water was injected into the MC and then removed and injected into the waveguide. Based on these water blanks, the LOD of the method was 0.69 Mm$^{-1}$, mainly due to the

uncertainty associated with the spectrometer measurement. The MC-LWCC LOD was higher than the LOD for the offline filter method at 0.10 Mm$^{-1}$ due to a smaller volume of air sampled by the MC. Alternatively, MC-LWCC blank variability can be estimated when sampling in clean background air when BrC levels are expected to be low. For example, the flight on 19 Aug 2019 did not encounter any smoke plumes, and the WS BrC was very low, together with other smoke tracers CO and BC. Using the time period from 19 Aug 2019 18:20:00 – 19:29:59 (CO =73 ppb and BC=0.5 ng/m$^3$) as the blank, the LOD ($3*\sigma$)

is calculated to be 1.53 Mm$^{-1}$. With this LOD, only ~40% of the data was above the LOD for the whole FIREX-AQ study period, below LOD periods were primarily when not sampling in smoke plumes (e.g., traveling to and from the fires and between fires). When the DC-8 was sampling smoke plumes (CO>300 ppbv), which was the main focus in FIREX-AQ, more than 90% of the MC data was above the LOD, implying that this system was mainly useful for in-plume sampling.

The uncertainty of the MC-LWCC WS BrC system was calculated by propagating the uncertainties from air sampling volume,

liquid extraction volume, the spectrometer absorption measurement, baseline drift correction, and hysteresis correction. Besides the hysteresis correction, the relative uncertainty for other components was approximately 5%. The combined overall estimated uncertainty based on these variables was roughly 28%.

***PILS-LWCC:*** Each PILS-LWCC system acquired continuous data at 10 s (NOAA PILS-LWCC) or 16 s resolution (CSU PILS-LWCC). The measurement precision can be determined for in-flight measurements of ambient air with no detectable

BrC absorption. Figure 9 shows an Allan deviation (Werle et al., 1993; Allan, 2016) plot for Abs$_{365nm}$ calculated for the CSU PILS-LWCC (1 h 38 m period on 13 Aug 2018) and NOAA PILS-LWCC (2 h 3 m period on 11 Aug 2019), during measurements in background air. The 1σ Allan Deviation for 30 s averaging is 0.29 Mm$^{-1}$ and 0.008 Mm$^{-1}$ for CSU and NOAA PILS-LWCC respectively, equivalent to 3σ LODs of 0.89 Mm$^{-1}$ and 0.03 Mm$^{-1}$, respectively. Although the NOAA PILS-LWCC shows high precision over short intervals, the Allan deviation plot indicates instrumental drift at all time scales.



The uncertainty of the CSU PILS-LWCC system was estimated to be 12% based on the uncertainties in the air and liquid flow rates and the ambient and background absorption measurement. The uncertainty of the NOAA PILS-LWCC system was calculated to be 11%.

### 3.6 Recommendations for further improvements

The MC-LWCC and PILS-LWCC instruments successfully measured WS BrC throughout the WE-CAN and FIREX-AQ
aircraft deployments. However, future improvements have the potential to improve the detection limits and time response of these instruments.

For the MC-LWCC, the cleaning procedure between consecutive samples was not sufficient and resulted in a large hysteresis effect (44%) when moving in and out of smoke plumes. This could be improved by introducing a clean air flow through the MC to wash the interior wetted surfaces between samples. Shortening the tubing running between the MC and LWCC could
reduce the dead volume, improving the time response and hysteresis. Arranging the MC with a minimal distance between the sample extraction port and the MC bottom would decrease the residual liquid sample within the MC (or by placing the MC sample port on the bottom of the MC body). The addition of an automated filter at the sample inlet would allow repeated blank measurements of filtered air, and a better assessment of the LOD. This would also be useful in quantifying measurement hysteresis.
Increasing the sensitivity of the MC system would allow obtaining more data, especially outside of the smoke plumes. The extraction volume and sampling flow rate cannot be significantly altered. The most effective way to increase sensitivity is to increase the sampling time, but the time resolution would decrease. Although not ideal for this study due to the fast speed of the DC-8, this could be a viable solution for ground-based studies, where decreasing the time resolution from 2.5 min to 30 or 60 min may still produce acceptable time resolved data, with 12 to 24 times improvement in the sensitivity (lower LOD). This
higher sensitivity could also reduce measurement uncertainty.

For the PILS-LWCC time response could be improved and hysteresis reduced by using a peristaltic pump instead of syringe pumps to move the liquid sample from the PILS to the LWCC, and by decreasing the volume between the impaction plate to the LWCC and any places where liquid gets stalled in the system. This would entail use of short small-bore tubing, and the smallest internal volume possible for the debubbler (e.g., use of a inline bubble trap with PTFE membrane with vacuum assist)
and liquid filter, and reducing the dead volume in the syringe pumps (i.e. using a cone tip instead of a flat tip on the piston).

For all liquid systems, adding surfactants to the water to reduce the water surface tension can reduce hysteresis (Rastogi et al., 2009), but must be selected so as not to interfere with the analytical systems. It is known that introduction of air bubbles in the sample line, that are completely removed just upstream of the LWCC, can reduce hysteresis by "wiping" the walls of the wetted surfaces as they pass through the liquid system.
Drift in the light source intensity may contribute to instrument drift, seen in the Allan Deviation plot in Fig. 9. This could be improved by temperature-controlling the light source and monitoring its output intensity. The pressure-controlled inlet system for the NOAA-PILS also seems to have distinct advantages since it allowed the use of the peristaltic pumping system instead





of syringe pumps and likely dampens other variability in the complete PILS-LWCC due to changes in ambient pressure with aircraft altitude changes and possible turbulence in the inlet and air sample lines.

**4. Conclusion**

We present a comparison of three WS BrC measuring systems, including two PILS-LWCC and a newly developed MC-LWCC. The new system was based on expanding the analytical capabilities of a mist chamber (MC) sampling system on the NASA DC-8 research aircraft which had been extensively used in past studies for measuring inorganic gases and aerosol particles. The new system was deployed during the NASA FIREX-AQ and contrasted with the performance of the PILS-

LWCC systems for measuring WS BrC on the NSF C-130 aircraft as part of WE-CAN and on the NOAA Twin Otter during FIREX-AQ. These three systems used almost identical BrC analytical methods (LWCCs and spectrophotometers) to determine levels of light absorbing chromophores in liquid water samples, whereas the particle collection and liquid handling systems differed. Using a dual MC system operating in batch mode, the MC-LWCC measurement time resolution was 2.5 min. Sampling air at 50 SLPM with 12 mL of collection water and a 2.5 m long LWCC, the LOD of the system for measuring WS

BrC was 1.53 Mm$^{-1}$ with an estimated 28% uncertainty. For comparison the filter sampling system with offline analysis of BrC using an identical LWCC and spectrophotometer had a LOD of 0.10 Mm$^{-1}$ and uncertainty of 16%. The CSU PILS-LWCC system sampling at an air flow rate of 15 SLPM had a LOD of 0.89 Mm$^{-1}$ and uncertainty of 12%, and NOAA PILS-LWCC with 6 SLPM had a LOD of 0.03 Mm$^{-1}$ and uncertainty of 11%, both operating as continuous samplers. Spectral drift due to air bubbles in the sample line and BC that passed the liquid particle filter was an issue with the MC-LWCC, but not as apparent

in the PILS-LWCC systems, and had no effect on the filter sampling method. Hysteresis (smearing) of samples between consecutive measurements was a major artifact in this study for the MC-LWCC and on many occasions for the CSU PILS-LWCC, which was clearly seen in this study of smoke plumes measured near the fires. For the MC-LWCC, the hysteresis was largely due to not completely flushing the MC with clean water and not generating a mist to wash all internal surfaces, with a minor contribution from hysteresis in the sample lines. For the CSU PILS-LWCC, hysteresis issues were due to the size of the

wetted area of the impaction plate (which was small, a specific design feature of the system) and the liquid flow system that included the debubbler, liquid filter, syringe pumps, and sample lines. A hysteresis correction results in sharper changes in concentrations, that more closely track changes in CO when transitioning from sampling in and out of wildfire smoke plumes with the aircraft (average aircraft speed for NASA DC-8=200 m/s and for NSF C-130=100 m/s), and increases the in-plume BrC levels, but produced more variability (noise) in the CSU PILS-LWCC dataset. The NOAA PILS-LWCC showed very

little evidence of sample hysteresis possibly due to the different liquid sample flow system that used a peristaltic versus syringe pumps (e.g., for each stroke first liquid into pump is last out) to move the sample liquid, and the overall lower concentrated plumes encountered. For the MC-LWCC, the online WS BrC data was in good agreement with the offline WS BrC measured with filters with a regression slope of 0.91 and R$^2$=0.89. Since the MC-LWCC should not be susceptible to WS BrC volatility artifacts known to occur in filter sampling, the good agreement suggests that there are few artifacts associated with the filter



method and that much of the BrC was likely not highly volatile. In this study, the MC-LWCC was only of sufficient sensitivity to measure BrC levels in smoke plumes, the filter sampling system with much higher mass loading (~20 times higher) could measure WS BrC even in continental background conditions. As this was the first attempt at WS BrC measurements with a mist chamber, possible improvements to the MC-LWCC system were proposed.

**Acknowledgements:** This work was supported by NASA grant 80NSSC18K0662, NSF grant AGS-1650786, and NOAA grant NA17OAR4310010. The authors thank Delphine Farmer, Sonia Kreidenweis and Paul DeMott for the WE-CAN SP-2 data, and Glenn Diskin for CO and Joshua Schwarz for BC FIREX data. We also thank all pilots and crew of the NASA DC-8, NSF C-130, and NOAA Twin Otter.

**Data Availability:**

FIREX-AQ data can be downloaded from the NOAA/NASA FIREX-AQ data archive: FIREX-AQ DOI: 10.5067/SUBORBITAL/FIREXAQ2019/DATA001. WE-CAN data can be found at https://data.eol.ucar.edu/project/WE-CAN.

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



**Tables**

**Table 1. Overview of BrC instruments deployed during WE-CAN 2018 and FIREX-AQ 2019**

| | MC-LWCC | CSU PILS-LWCC | NOAA PILS-LWCC |
|---|---|---|---|
| Research Institution | University of New Hampshire Georgia Institute of Technology | Colorado State University | NOAA Chemical Sciences Laboratory |
| Field Campaign | FIREX-AQ 2019 | WE-CAN 2018 | FIREX-AQ 2019 |
| Aircraft | NASA DC-8 | NSF C-130 | NOAA Twin Otter |
| Aircraft Altitude | 200 – 1000 hPa | 430 – 1000 hPa | 630 – 1000 hPa |
| Inlet Pressure Control | None | None | 620 hPa |
| Aerosol Collection | Mist chamber (Scheuer et al., 2003) | Particle-into-Liquid Sampler | Particle-into-Liquid Sampler (PILS-4001, Brechtel) |
| Liquid Transfer | Two syringe pumps | Syringe pumps and a peristaltic pump | Peristaltic pump |
| Solvent | Water | Water | Water |
| Aerosol Collection Efficiency and System Dilution | Standard addition of trifluoracetic acid to determine evaporative loss in mist chamber | Dilution ratio due to condensation was obtained from previous measurements | aerosol collection efficiency of 0.8 for $D_P < 1$ µm from calibration with aerosolized sucrose particles post-campaign |
| Removal of Gas-Phase VOCs | N.A. | Parallel plate carbon filter denuder | Parallel plate carbon filter denuder (DN-100, Sunset Laboratory) |
| Light Source | Deuterium and halogen lamps (DH-Mini, Ocean Optics) | Deuterium and halogen lamps (DH-Mini, Ocean Optics) | Deuterium and halogen lamps (DH-Mini, Ocean Optics) |
| Liquid Waveguide | 2.5 m (LWCC-3250, World Precision Inst.) | 2.5 m (LWCC-3250, World Precision Inst.) | 2.5 m (LWCC-3250, World Precision Inst.) |
| Spectrometer | Ocean Optics FLAME-T-UV-VIS | Ocean Optics FLAME-T-UV-VIS | Ocean Optics QE Pro |
| Spectral Range | 300 - 700 nm | 300 - 700 nm | 309 - 682 nm |
| Spectral Resolution | 1.4 nm | 1.4 nm | 3.3 nm |
| Zero Measurement | N.A. | Filtered air measured for 10 min manually twice per flight | Filtered air measured for 6 min every 1.5 h |





| Detection Limit (3σ) at 365 nm | 1.53 Mm⁻¹ | 0.89 Mm⁻¹ | 0.03 Mm⁻¹ |
|---|---|---|---|
| Uncertainty | 28% | 12% | 11% |





**Figures**

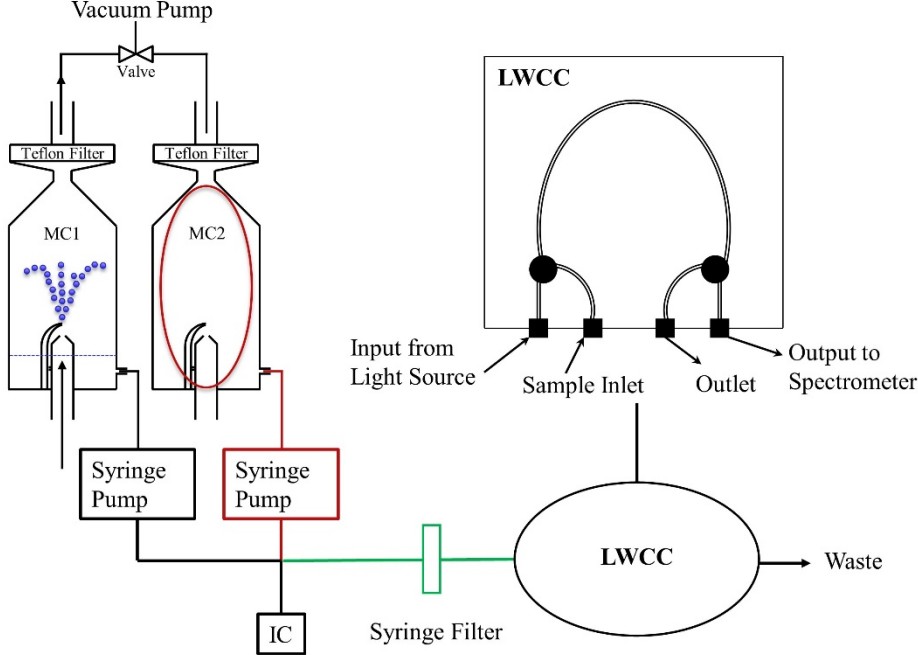

**Figure 1. Flow diagram of the MC-LWCC instrument for WS BrC. Blue dots represent the mist generated in the scrubber. Red lines are the first hysteresis components described in Section 3.3, and green lines are the second components.**




**Figure 2. Flow diagrams of the (a) CSU PILS-LWCC and (b) NOAA PILS-LWCC instruments for WS BrC. For the CSU PILS-LWCC, a combination of syringe and peristaltic pumps was used for handling the liquid flows. For the NOAA PILS-LWCC, a peristaltic pump was used for all flows.**





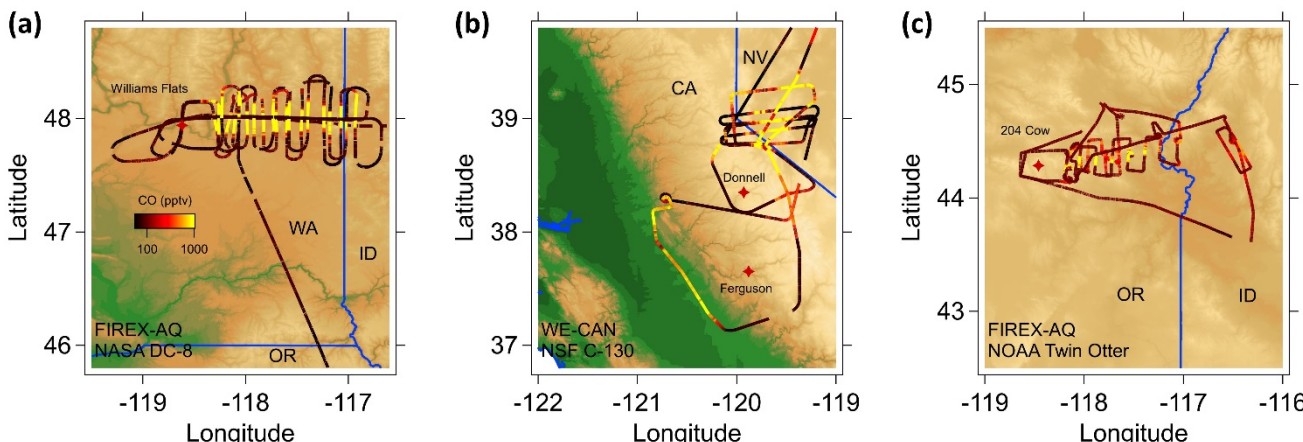

**Figure 3. Examples of flight tracks for measurements made near and downwind of fires in the western USA for (a) the NASA DC-8 on 7 Aug 2019 in FIREX-AQ (MC-LWCC); (b) the NSF C-130 on 6 Aug 2018 in WE-CAN (CSU PILS-LWCC); (c) the NOAA Twin Otter on 24 Aug 2019 in FIREX-AQ (NOAA PILS-LWCC). Each flight is color coded by the CO mixing ratio. Fires are labelled as red stars.**





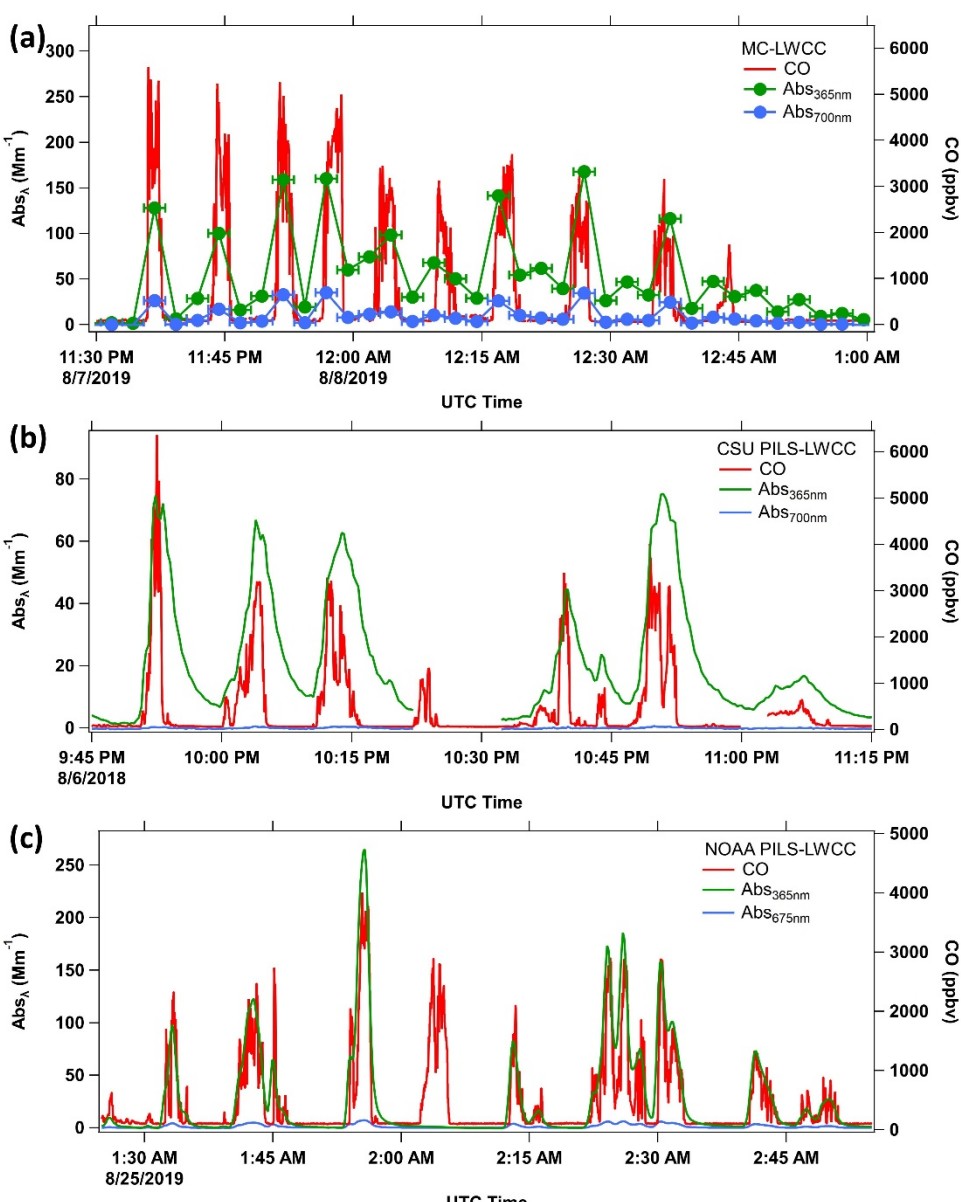

**Figure 4. Example data from sampling in smoke plumes by MC-LWCC and PILS-LWCC (CSU and NOAA) systems for the flights shown in Figure 3. Time series of Abs$_{365nm}$ (green), Abs$_{700nm}$ or Abs$_{675nm}$ (blue), and CO (red) for the (a) FIREX-AQ NASA DC-8 flight on 7 Aug 2019, (b) WE-CAN NSF C-130 flight on 6 Aug 2018, and (c) FIREX-AQ NOAA Twin Otter flight on 24 Aug 2019. The sampling frequencies were (a) MC-LWCC 2.5 min; (b) CSU PILS-LWCC 16 s; (c) NOAA PILS-LWCC 10s; and CO 1 s. Horizontal error bars in (a) represent the MC sampling interval.**




Atmospheric Measurement Techniques Discussions - Open Access EGU

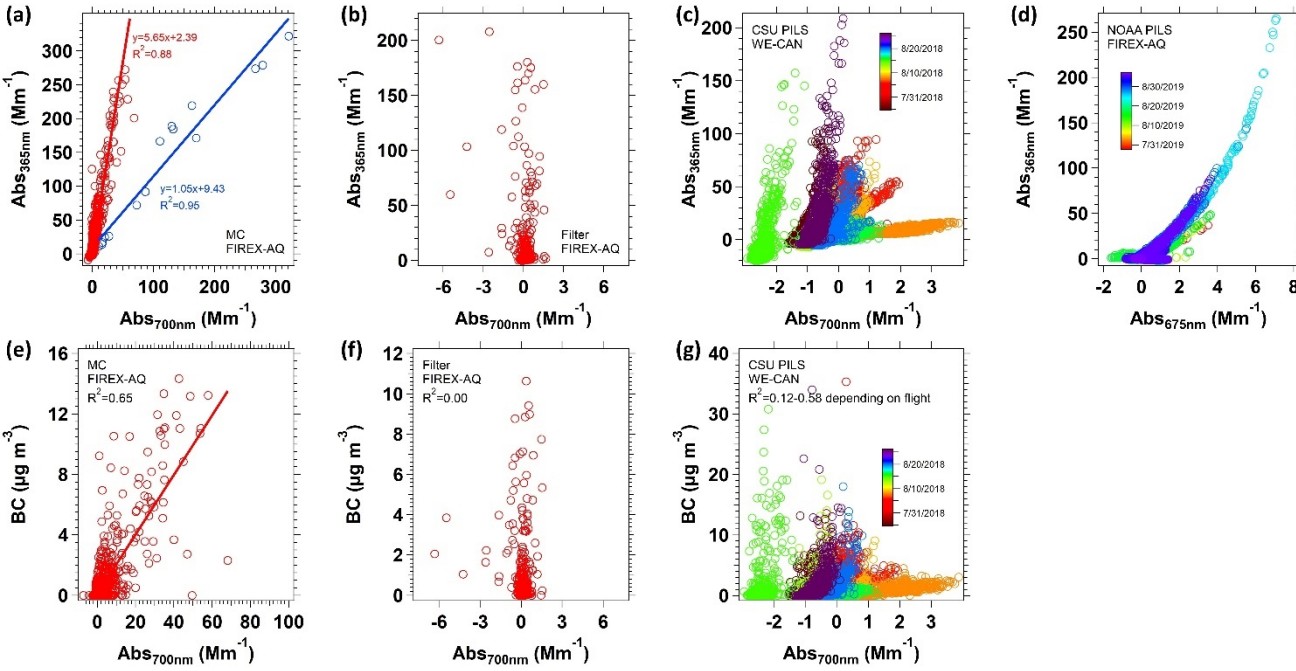

**Figure 5. Scatter plots between Abs$_{365nm}$ and baseline (Abs$_{700nm}$; Abs$_{675nm}$ for NOAA PILS) with (a) MC, (b) filter, (c) CSU PILS, and (d) NOAA PILS. The corresponding scatter plots between Abs$_{700nm}$ and BC are shown in (e, f, and g). For the (a) MC-LWCC system, two groups of data are classified visually and fit with an orthogonal distance regression. Red data is the baseline drift due to BC passing the particle filter, and blue data is because of small air bubbles in the LWCC. For PILS-LWCC (c and d), data are color coded by sampling date.**





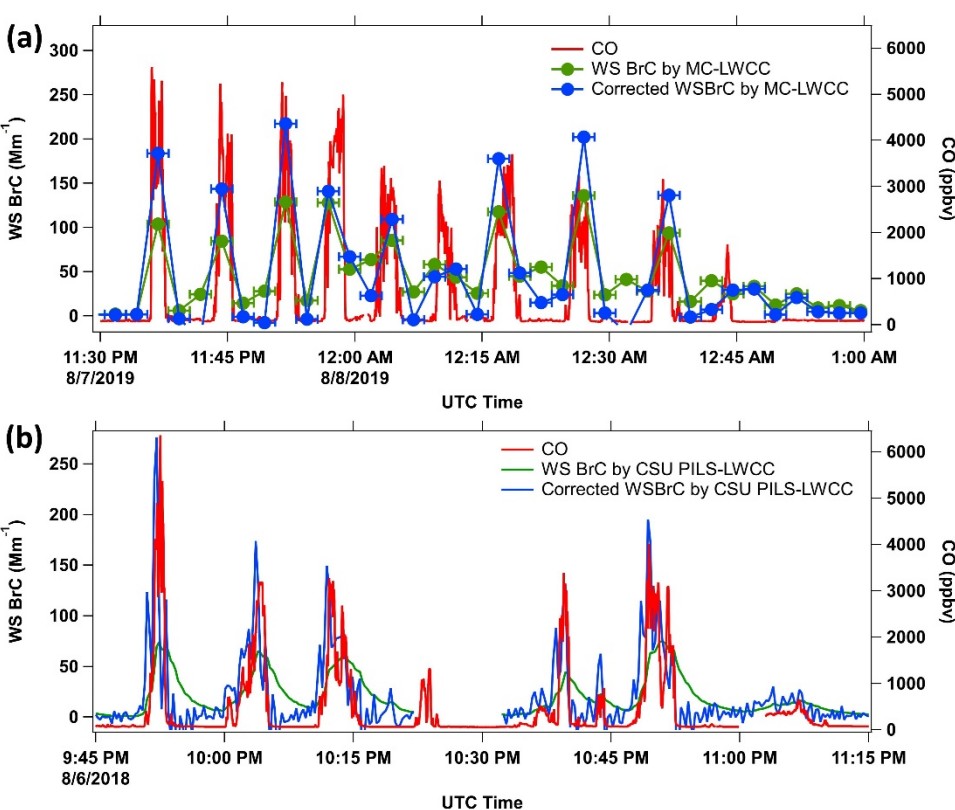

**Figure 6.** Time series of water-soluble BrC (WS BrC) corrected for baseline drift and hysteresis (blue) compared to original data (green) and CO concentrations (red) for (a) the MC-LWCC measurement and (b) the CSU PILS-LWCC measurement for the same flights shown in Figure 4a and 4b, respectively. Horizontal error bars in (a) represent the MC-LWCC interval.




**Figure 7. Scatter plots of WS BrC versus CO concentration for the MC-LWCC before (a) and after the (b) hysteresis correction, and for the CSU PILS-LWCC before (c) and after the (d) hysteresis correction.**




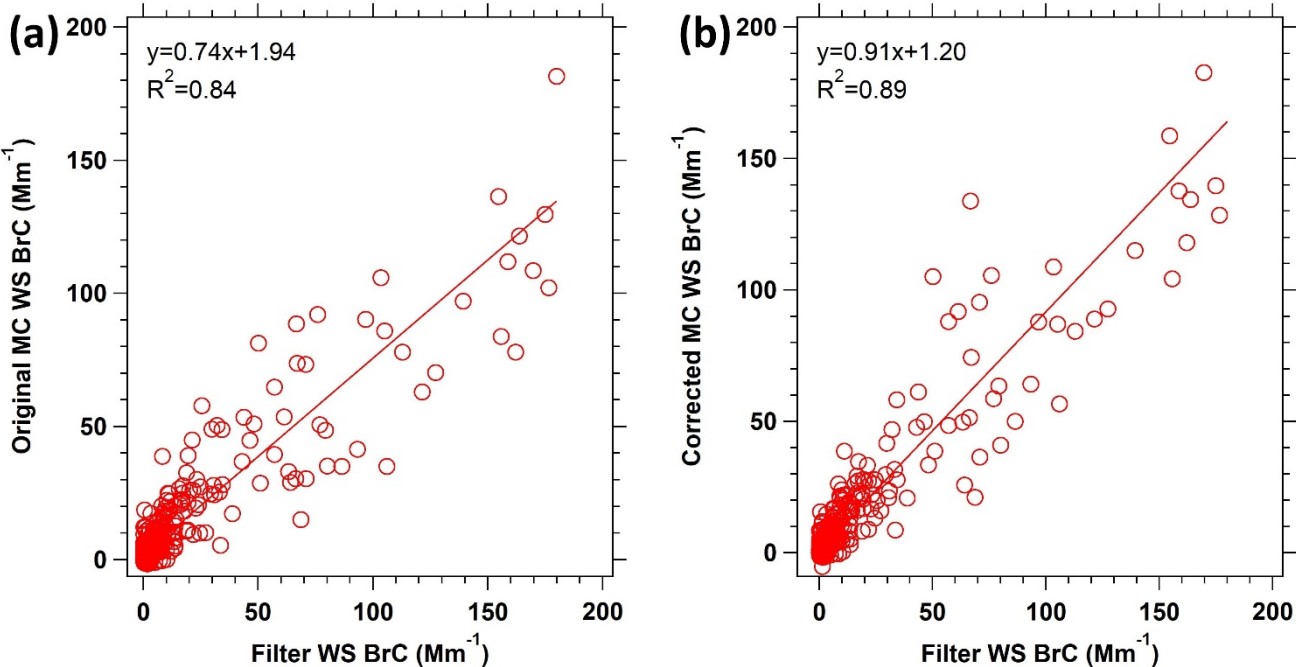


**Figure 8. Comparison of the WS BrC determined online using the MC-LWCC vs. offline from extraction of filters. (a) shows the original MC-LWCC WS BrC data, and (b) shows the MC data with the hysteresis correction applied. The data is fitted by orthogonal distance regression.**





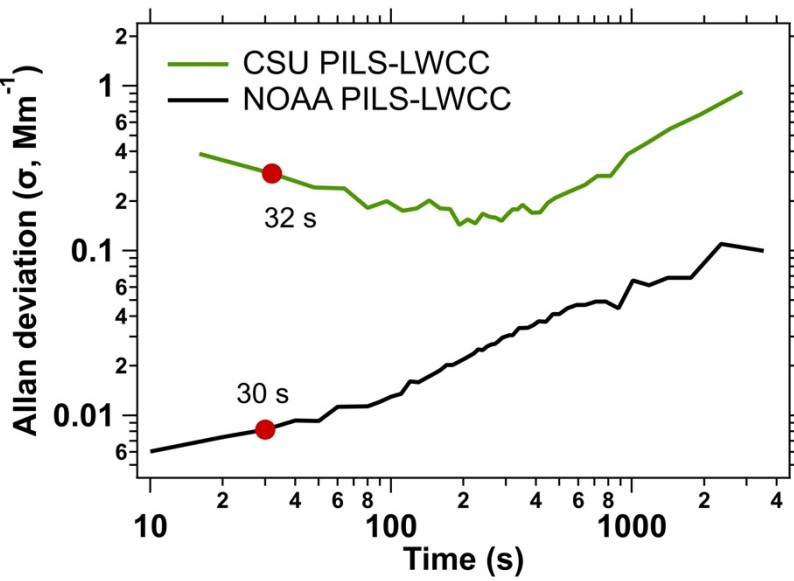


**Figure 9. Allan Deviation plot calculated for all WE-CAN data without plumes on Flight 10 (1:38h from 13 Aug 2018 19:16:29-20:54:58) and FIREX-AQ NOAA Twin Otter flight data 11 Aug 2019 flight (2:03h from 11 Aug 2019 22:46:04 - 12 Aug 2019 00:49:54). There were no plumes during either of these periods. Data in red dots are used to determined LODs for both systems.**