# Peer review of "Assessment of online water-soluble brown carbon measuring systems for aircraft sampling"

_Atmospheric Measurement Techniques, 2021_

## Author Comment (AC1)

Response to Reviewer 1 comments

The authors thank the reviewer for the positive overview response to the manuscript and constructive comments. *We have responded below and believe these changes have improved upon the paper. (Lines numbers below refer to the original submitted version of the manuscript with tracked change (All Markup) unless otherwise noted).*

Reviewer #1

The manuscript presents are carefully conducted assessment of instruments for collecting and measuring water-soluble brown carbon (BrC) in atmospheric aerosol samples from airborne platforms. The investigated methods for sampling water-soluble BrC particles include two Particle-into-Liquid-Samplers (PILS) and one Mist Chamber (MC). All particle samplers transfer the collected BrC particles to a liquid waveguide capillary cell (LWCC) and grating spectrometer which measures the light absorption by the BrC- containing particles across the UV-VIS spectrum. The method assessment includes instrument deployments on different airborne platforms during different field experiments (NSF C-130 aircraft during WE-CAN 2018; NASA DC-8 and NOAA Twin Otter aircraft during FIREX-AQ 2019), where the instruments sampled fresh and moderately aged wildfire plumes. The assessment reports method-characteristic parameters like limit of detection and uncertainty.

The study is carefully designed and performed. The presentation of the results is well structured and clear. The manuscript fits well into the scope of the journal and can be accepted for publication, after few minor revisions have been implemented.

SPECIFIC COMMENTS

1. The manuscript assesses the measurement of light absorption by water-soluble BrC. In the manuscript, the used nomenclature refers to brown carbon measurements in terms of light absorption coefficients which are reported in $Mm^{-1}$, whereas black carbon is referred to in mass concentration units. For consistency, a clearer term for BrC light absorption may be used. I am aware that the text may become a little more cumbersome, but in the current version it is confusing to switch between BrC in units of $Mm^{-1}$ and BC in units of $\mu g\ m^{-3}$.

   **Response:** *We acknowledge that different units of BrC and BC may lead to confusion. Currently, BrC is always derived from measurements of light attenuation, which has the units of $Mm^{-1}$. BC can be measured by a variety of methods that report BC levels in different units, depending on the method. These include light attenuation (e.g., various aethalometers, units $Mm^{-1}$), photoacoustic measurements of particle heating when irradiated (e.g., photoaccoustic absorption spectroscopy (PAS), units of $Mm^{-1}$) and the incandescence signal produced from a particle irradiated with light (e.g., SP2, units of $ug/m^3$). Since BrC measurements in this paper are limited to light absorption methods, the mass absorption cross-section (MAC, which is the light absorption coefficient/mass of BrC) is needed to convert to mass. There is no unique MAC value for BrC at a given wavelength since BrC is composed of a host of individual chromophores, each with their own MAC, whose contribution to the overall particle MAC varies with emissions and particle atmospheric aging. To address this*

*reviewer's question, we have added "the light absorption of BrC" in Line 121, Line 143, and "BrC absorption coefficient" Line 147, which are overviews of this work. We have also added "BrC is reported in the form of light attenuation (units of Mm$^{-1}$) and not converted to a mass concentration because there is no constant BrC mass absorption cross-section (MAC) value at a given wavelength as BrC is composed of multiple chromophores that change with emissions and atmospheric evolution." To Line 128.*

2. In section 2.6, the authors point to the fact that the reported light absorption measured in the liquid phase is not identical to the light absorption which would be measured in the airborne state of the particles because of size-dependent effects related to Mie theory. A short paragraph would help to explain whether the wavelength-dependence of light absorption coefficients measured in the liquid phase deviate from the respective properties measured in the gas phase, since absorption Ångström exponents (AAE) are used in the manuscript.

**Response:** *In this paper, we have presented solution-phase absorption, and have not applied Mie theory to calculate the particle-phase absorption. Our previous work (Liu et al., 2013) has described the Mie correction and equations in detail. Those references are included in Section 2.6, but we prefer to leave the details out of this paper, since that method is not used here.*

*The AAEs in the liquid phase and the aerosol phase are not directly comparable, because the Mie theory correction is wavelength-dependent. Moosmüller et al. (2011) have shown that the AAE in a bulk solution is comparable with AAE for modelled small spherical particles, but the modelled AAE is smaller than bulk solution AAE when particle size increases, especially when the wavelength range is larger than 300nm.*

3. In the discussion of Fig. 5a (line 351 and following), a short section on the meaning of the red symbols would help. The use of these values is mentioned in the paragraph on the baseline correction (line 379) but mentioning their meaning earlier would increase the clarity of presentation.

**Response:** *"Red data points are typical BrC measurements that have high absorption at short wavelength, but insignificant absorption at long wavelength, and $Abs_{700nm}$ is possibly due to the penetration of small BC particles through the liquid filter." has been inserted to Line 361 for clarity.*

4. In line 428, the authors refer to the assumptions used for the decomposition of the hysteresis effect. One short sentence on the explanation of the meaning of the second assumption would clarify the presentation.

**Response:** *The sentence in Line 440 has been changed to "The left side of the equations are zero due to the second assumption that WS BrC absorption is negligible outside the plume."*

MINOR ISSUES

1. In the Introduction, the integrating sphere method for separating BC from BrC is missing; see Wonaschütz et al., (2009). It would be worth considering to include this reference.

**Response:** *A brief introduction of ISM method have been added to Line 74 "Another BrC separation method employing an Integrating Sphere Method, introduced by Wonaschütz et al. (2009), first assumes all absorption at long wavelength (660 nm) is due solely to BC, then an iterative technique is used to obtain BrC absorption based on calibration curves from simulated BC and BrC (carbon black and humic acid salt). The iteration can account for BrC absorption at longer wavelength, but there is difficulty in obtaining a calibration line for real ambient samples.".*

2. The effect of coating on aerosol light absorption properties was studied in-depth for urban and continental sites by Liu and co-workers (2015). This reference should also be considered in the introduction.

**Response:** Liu et al. (2015), Lack and Cappa (2010), and Lack et al. (2012) have been added to Line 72, and the text has been changed for clarity to be "*Studies that use an assumed AAE value introduce even greater uncertainty into the determination of BrC, since a range of values for BC from 0.6-1.9 has been observed due to the coating effect (Bergstrom et al., 2007; Lack and Cappa, 2010; Lack et al., 2012; Bond et al., 2013; Lan et al., 2013; Liu et al., 2015b; Li et al., 2016).*".

3. Line 172 ff: The description of the sampling sequence should also contain a brief description of MC2 and the connected IC analysis. Currently, the entire description is focusing on MC1 and the light absorption measurement.]

Response: *We have changed "MC1 was then ready for the next cycle of sampling" to be "MC1 is inactive until the start of the next sampling cycle. Once air sampling had begun by MC1, the liquid in MC2 was injected into the LWCC and IC, and then MC2 was cleaned." to Line 187.*

4. Line 176: Please correct "Absorbance" instead of "Aabsorbance".

**Response:** *The miss spelled word "Aabsorbance" has been corrected to be "Absorbance" in Line 184.*

5. The statement on authors' contributions should be added.

**Response:** *We have added "**Author Contribution:** RJW and JD provided the original idea of building the MC-LWCC. LZ and ES designed and built the MC-LWCC. AS was responsible for CSU PILS-LWCC and RAW was responsible for NOAA PILS-LWCC. TC, JK, EL, and MR made other measurements on the aircraft. LZ, AS, and RAW analyzed BrC data. LZ and RJW prepared the manuscript with contributions from all co-authors." to the end of text (Line 627).*

Reference

Lack, D. A., and Cappa, C. D.: Impact of brown and clear carbon on light absorption enhancement, single scatter albedo and absorption wavelength dependence of black carbon, Atmos. Chem. Phys., 10, 4207-4220, 10.5194/acp-10-4207-2010, 2010.

Lack, D. A., Langridge, J. M., Bahreini, R., Cappa, C. D., Middlebrook, A. M., and Schwarz, J. P.: Brown carbon and internal mixing in biomass burning particles, Proceedings of the National Academy of Sciences, 10.1073/pnas.1206575109, 2012.

Liu, J., Bergin, M., Guo, H., King, L., Kotra, N., Edgerton, E., and Weber, R. J.: Size-resolved measurements of brown carbon in water and methanol extracts and estimates of their contribution to ambient fine-particle light absorption, Atmos. Chem. Phys., 13, 12389-12404, 10.5194/acp-13-12389-2013, 2013.

Liu, S., Aiken, A. C., Gorkowski, K., Dubey, M. K., Cappa, C. D., Williams, L. R., Herndon, S. C., Massoli, P., Fortner, E. C., Chhabra, P. S., Brooks, W. A., Onasch, T. B., Jayne, J. T., Worsnop, D. R., China, S., Sharma, N., Mazzoleni, C., Xu, L., Ng, N. L., Liu, D., Allan, J. D., Lee, J. D., Fleming, Z. L., Mohr, C., Zotter, P., Szidat, S., and Prévôt, A. S. H.: Enhanced light absorption by mixed source black and brown carbon particles in UK winter, Nature Communications, 6, 8435, 10.1038/ncomms9435, 2015.

Moosmüller, H., Chakrabarty, R. K., Ehlers, K. M., and Arnott, W. P.: Absorption Ångström coefficient, brown carbon, and aerosols: basic concepts, bulk matter, and spherical particles, Atmos. Chem. Phys., 11, 1217-1225, 10.5194/acp-11-1217-2011, 2011.

Saleh, R.: From Measurements to Models: Toward Accurate Representation of Brown Carbon in Climate Calculations, Current Pollution Reports, 6, 90-104, 10.1007/s40726-020-00139-3, 2020.

Wonaschütz, A., Hitzenberger, R., Bauer, H., Pouresmaeil, P., Klatzer, B., Caseiro, A., and Puxbaum, H.: Application of the Integrating Sphere Method to Separate the Contributions of Brown and Black Carbon in Atmospheric Aerosols, Environmental Science & Technology, 43, 1141-1146, 10.1021/es8008503, 2009.

---

## Author Comment (AC2)

Response to Reviewer 2 comments

The authors thank the reviewer for the positive overview response to the manuscript and constructive comments. *We have responded below and believe these changes have improved upon the paper. (Lines numbers below refer to the original submitted version of the manuscript with tracked change (All Markup) unless otherwise noted).*

Brown carbon (BrC) plays an important role in climate and atmospheric chemistry, but determining the mass concentration and absorption of BrC is still challenging. This manuscript reports the first direct, aircraft-based online measurements of water-soluble BrC in wildfire plumes by three methods based on liquid waveguide capillary cell and different aerosol collection techniques. The three methods are introduced in detail and a comprehensive evaluation of the measurement uncertainties are given. The authors also established new algorithms for the correction of hysteresis effect owing to the retention of liquid on the internal components of the system. This study provides a good example of online water-soluble BrC measurements and is of great value for similar measurement in the future. I therefore recommend the publication of the manuscript on AMT. I only have some minor comments as list below:

1. L357: The presence of Abs_700nm is attributed to BC particles passing through the filter (diameter<0.22 um). Since there is SP2 measurement in parallel, it is possible to have an estimate of the BC mass concentration in particles smaller than 0.22 um. Is it true that a higher R2 will be obtained for the correlation between Abs_700nm and the mass concentration of tiny BC?

**Response:** *We have used BC from SP2 and size distribution measured by a Laser Aerosol Spectrometer (LAS) to estimate the BC mass under 0.22 um in FIREX-AQ (the DC-8), but the $R^2$ did not improve (new $R^2=0.57$). In WE-CAN, the size distribution was measured by a Scanning Mobility Particle Sizer (SMPS), and the new $R^2$ ranged between 0.14-0.58. The main reason the $R^2$ does not increase is that the hysteresis effect has not been removed at this step, so in some cases, large Abs700nm was observed, but only small BC was observed.*

2. L375: I think the equation given in L379 is always correct assuming an AAE_BC=1. Why do the authors use a simplified equation (L381) with larger overestimation for the correction of CSU PILS-LWCC?

**Response:** *As can be seen in Figure 5c, the slope of Abs365nm to Abs700nm is larger than ~25, except for data points in orange, which have a slope of ~6. The simplified method will only overestimate BrC by less than 4%. We are reporting 12% uncertainty for the BrC absorption with the CSU PILS-LWCC, so 4% isn't necessarily negligible. Additionally, the absorption observed at 700 nm may also be due to the insoluble S-BrC, which has been proposed by Saleh (2020), but we consider this class of BrC as insoluble BC.*

3. L415: The statement is a bit confusing, seems not consistent with Eq. 4.

**Response:** *The sentence in Line 427 has been changed to "We assume the observed WS BrC absorption at the i-th sample is due to a% of the real WS BrC during the time period of the i-th sample, b% due to (i-1)-th sample from the tubing, and c% due to (i-2)-th sample from the MC."*

4. L390: The correction of hysteresis effect seems to strongly rely on the contrast between measurement in plume with high BrC concentration and in background with nearly no BrC. Is the method also suitable for the correction of BrC measurements with much lower temporal variability in BrC concentration?

**Response:** *Yes, the correction method provided in the manuscript is suitable for other measurements with lower variability in BrC concentration, but new coefficients may need to be obtained. Additionally, it is hard to find a sharp cut when the aircraft transitions between smoke to background air when the CO or BrC has less variability, and the hysteresis effect would not be that obvious in a weaker plume. In FIREX-AQ, we did not encounter any good plume transects that had a sharp cut exiting the plume with CO less than 1000 ppbv to obtain these coefficients. However, these coefficients obtained from plumes with CO mixing ratios ranging from 1500 ppbv to 5000 ppbv in this study did not change significantly, implying other factors (tubing length and mist chamber shape) have a stronger effect on these coefficients than the concentration.*

5. Fig. 4C: At 2:05 there is a strong peak of CO. I am wondering why there is no BrC measured.

**Response:** *Thanks for pointing out this mistake. There is a data gap from 8/25/2019 02:00:23 to 8/25/2019 02:08:05 explaining the cause of higher CO, but no BrC in the original plot. Figure 4 has been modified to correct this.*